# Distinct B-Cell Specific Transcriptional Contexts of the BCL2 Oncogene Impact Pre-Malignant Development in Mouse Models

**DOI:** 10.3390/cancers14215337

**Published:** 2022-10-29

**Authors:** Lina Zawil, Tiffany Marchiol, Baptiste Brauge, Alexis Saintamand, Claire Carrion, Elise Dessauge, Christelle Oblet, Sandrine Le Noir, Frédéric Mourcin, Mylène Brousse, Paco Derouault, Mehdi Alizadeh, Yolla El Makhour, Céline Monvoisin, Julien Saint-Vanne, Simon Léonard, Stéphanie Durand-Panteix, Karin Tarte, Michel Cogné

**Affiliations:** 1Immunology Department, Faculty of Medicine, Limoges University, Cnrs Umr 7276, Inserm U1262, 2 Rue du Dr Marcland, 87000 Limoges, France; 2UMR U 1236, Univ Rennes 1, INSERM, EFS Bretagne, Equipe Labellisée Ligue Contre le Cancer, 35000 Rennes, France; 3Chu Dupuytren, 87000 Limoges, France; 4Immunology Department, Science Faculty, Lebanese University, Beirut P.O. Box 6573/14, Lebanon; 5Siti Laboratory, Chu Rennes, 35000 Rennes, France; 6LabEx IGO “Immunotherapy, Graft, Oncology”, 44000 Nantes, France

**Keywords:** oncogene deregulation, translocation, lymphoma, plasmacytosis, germinal center

## Abstract

**Simple Summary:**

Beyond the classical t(14;18) translocation associated with follicular lymphoma, *BCL2* is deregulated in multiple B-cell malignancies, including some cases of myeloma, and through diverse genetic anomalies. It is currently unclear how the various deregulation patterns mechanistically impact the phenotype of theses malignancies. We designed two different BCL2 deregulation models in transgenic mice, whereby the oncogene was either associated with the IgH3′RR superenhancer, as in t(14;18), or inserted into the kappa light chain locus. We compared the impact of these models on B-cell fate and lymphoid tissues. Linkage to the IgH superenhancer showed a quite specific impact on germinal center B cell populations. The Ig kappa model was much less specific and strongly boosted the plasma cell in-flow and the accumulation of long-lived plasma cells.

**Abstract:**

Upregulated expression of the anti-apoptotic *BCL2* oncogene is a common feature of various types of B-cell malignancies, from lymphoma to leukemia or myeloma. It is currently unclear how the various patterns of deregulation observed in pathology eventually impact the phenotype of malignant B cells and their microenvironment. Follicular lymphoma (FL) is the most common non-Hodgkin lymphoma arising from malignant germinal center (GC) B-cells, and its major hallmark is the t(14:18) translocation occurring in B cell progenitors and placing the *BCL2* gene under the control of the immunoglobulin heavy chain locus regulatory region (IgH 3′RR), thus exposing it to constitutive expression and hypermutation. Translocation of *BCL2* onto Ig light chain genes, *BCL2* gene amplification, and other mechanisms yielding *BCL2* over-expression are, in contrast, rare in FL and rather promote other types of B-cell lymphoma, leukemia, or multiple myeloma. In order to assess the impact of distinct BCL2 deregulation patterns on B-cell fate, two mouse models were designed that associated *BCL2* and its full P1-P2 promoter region to either the IgH 3′RR, within a “3′RR-*BCL2*” transgene mimicking the situation seen in FL, or an Ig light chain locus context, through knock-in insertion at the Igκ locus (“Igκ-*BCL2*” model). While linkage to the IgH 3′ RR mostly yielded expression in GC B-cells, the Igκ-driven up-regulation culminated in plasmablasts and plasma cells, boosting the plasma cell in-flow and the accumulation of long-lived plasma cells. These data demonstrate that the timing and level of BCL2 deregulation are crucial for the behavior of B cells inside GC, an observation that could strongly impact the lymphomagenesis process triggered by secondary genetic hits.

## 1. Introduction

Follicular lymphoma (FL) is the most frequent indolent non-Hodgkin’s lymphoma (NHL), representing about 25% of B-cell malignancies [1]. This germinal center (GC) B-cell malignancy results from the malignant and clonal accumulation of centrocytes and follows a multistep lymphomagenesis process evolving over decades before clinical manifestations. After a long premalignant phase, the clinical course is slow and characterized by multiple relapses associated with increased resistance to therapy. Over time, approximately 30% of FL cases transform into aggressive diffuse large B-cell lymphoma (DLBCL).

The major FL genetic hallmark is the t(14;18) (q32, q21) translocation, which is found in 90% of FL cases and deregulates the anti-apoptotic gene *BCL2* by displacing it under the control of immunoglobulin (Ig) heavy chain locus regulatory elements. A less frequent translocation, t(3;14), involves BCL6 and hereby indirectly deregulates BCL2 [2]. Contrasting to MYC in Burkitt lymphoma, for which variant translocations onto Ig light chain loci are common, *BCL2* t(2;18) and t(18;22) variant translocations to light chain genes are rare in FL but found in 9% of chronic lymphocytic leukemia cases [3,4]. In addition to translocations, chromosomal amplification of *BCL2* is also observed in DLBCL and in mantle cell lymphoma [5]. The diverse patterns of BCL2 deregulation are thus likely impacting the phenotype of BCL2-driven lymphoproliferative disorders.

The t(14;18) translocation stands as an aberrant V(D)J recombination product, joining double-strand breaks (DSBs) induced by RAG both in the IgH JH region (at position 14q32) and close to *BCL2* oncogene (at position 18q21) and hereby imposing an IgH-like pattern on *BCL2* gene accessibility and transcription [6]. BCL2 is a transmembrane mitochondrial protein with dual roles, both inhibiting apoptosis and cell cycle entry. In mature B cells, expression of the IgH-translocated *BCL2* gene is mostly under the control of the transcriptional enhancers from the IgH locus 3′ regulatory region (3′RR), which are the main locus drivers at mature B-cell stages [7]. These enhancers notably ensure accessibility of the locus to class switch recombination (CSR), somatic hypermutation (SHM), and hyper-transcription in activated B cells and plasma cells (PCs) [7,8,9,10]. The functional result of this translocation is the imbalance between cell survival and death at a stage where mature B cells are intensely exposed to affinity-based selection within the GC. Normal GC B-cells are indeed prone to undergo cell death unless they are positively selected and induced to enter the memory B cell or the plasma cell compartments [11,12].

BCL2 is normally repressed in the GC to ensure the selection of high-affinity B cells, while apoptosis is promoted by proteins such as BAK, BAX, and BAD. BCL2 deregulation yielded by t(14;18) thus jeopardizes the GC B cell survival checkpoint, and *BCL2* gene translocation stands as an early driver hit initiating lymphomagenesis in FL. Among naive B cells that have left the bone marrow (BM) and are circulating in the periphery, those harboring the t(14;18) translocation display a selective advantage during the GC reaction allowing them to persist as atypical memory B cells already carrying features of FL-like cells (FLLCs). Upon future Ag encounter, these premalignant FLLCs are prone to re-enter the GC and undergo additional SHM, with eventual additional oncogenic hits, under the iterative exposure to activation-induced deaminase (AID) [13,14,15]. Although an early and crucial anomaly marking pre-malignant B-cells, the t(14;18) is frequent in healthy adults and is not by itself sufficient for FL development, so the prevalence of FL does not exceed 0.03%. Among the most frequent additional hits, alterations of histone/chromatin modifying enzymes, including KMT2D, CREBBP, EZH2, and multiple linker-histone H1 family members, are collectively found in almost 100% of FL cases. Additional mutations, such as loss-of-function mutations of HVEM/TNFRSF14 or the introduction of N-glycosylation sites within Ig variable regions, have been shown to affect the crosstalk between tumor B cells and their surrounding microenvironment [15,16,17,18].

Exploring the mechanisms of such complex and multifactorial B-cell malignancies requires studying whole living organisms and has prompted the generation of several models in mice. Initial transgenic mouse models in which BCL2 expression was driven by the Eµ enhancer have mostly resulted in a polyclonal expansion of all B-cell compartments overexpressing BCL2, from progenitors to plasma cells [19,20]. Lymphoma development was reported mostly when the BCL2 deregulation was associated with other spontaneously selected or experimentally enforced genetic anomalies, notably involving Myc [21,22]. However, lymphoproliferation observed in such conditions rather involved immature B cells and did not provide models for post-GC low-grade human lymphoma. Strikingly, the most widely used transgenic mice considered as a pertinent model of human FL and based on the tumorigenic potential of BCL2 have been the vavP-*BCL2* transgenics, although they broadly overexpress BCL2 in all hematopoietic lineages, including T cells and thus do not recapitulate the natural story of human FL where BCL2 overexpression is restricted to mature B cells [23,24].

Therefore, in an aim to better understand the complex development of B-cell malignancies and the impact of B-cell specific *BCL2* deregulation, our current study explored two new mouse models, designed either for pan-B cell expression through a knock-in of *BCL2* in the Igκ locus or for specific targeting of activated B-cells and GC-B cells with a *BCL2* transgene driven by the IgH 3′RR enhancers. Since the 3′RR is a major driver of the IgH locus remodeling in the GC, the latter model is expected to mimic the BCL2 deregulation associated with FL more specifically. The *BCL2* promoter region has a characteristic structure comprising the P1 and P2 promoters, with P1 dominating normal lymphocytes, while a shift from P1 to P2 usage is observed in FL, in association with a strong increase in BCL2 expression within the GC. Contrary to previously reported *BCL2* transgenes, we thus included the full human *BCL2* promoter region in our *BCL2* gene cassettes. Indeed, this region includes transcription factor-binding sites, notably for the repressor BCL6 limiting *BCL2* transcription in normal GC cells [25]. The design of our constructs thus also aims at evaluating in mice the deregulation and the eventual mutations of the *BCL2* promoter region documented in patients.

On their own, both of our new mouse models promote mature B-cell expansion, but they, however, differ in terms of stage-specificity according to the context of *BCL2* deregulation, with 3′RR-*BCL2* mice showing the most GC-restricted phenotype. These models are pertinent for studying a pre-malignant stage in young mice, notably showing the impact of GC B-cell expansion on the other components of lymphoid tissues.

## 2. Materials and Methods

### 2.1. Cell Lines and Mouse Models

DoHH2, SU-DHL-4, SU-DHL-6, and OCI-Ly3 cell lines (Germinal center or activated B cell type Diffuse large B cell lymphoma (DLBCL)) were grown in RPMI 1640 medium supplemented with 10% Fetal calf serum (Dutscher, Catalog number: S1810-500) at 37 °C with 5% CO_2_ atmosphere).

All in vivo experiments were performed in accordance with animal ethical rules, and all protocols were authorized by the French Ministry of Research according to European Union regulations (APAFiS 13900). All mice were bred in a specific and opportunistic-free (SOPF) animal facility.

Two new mouse models were designed for this study (Figure 1A):The “3′RR-*BCL2*” model, with random integration of a transgene that contains a human *BCL2* gene cassette driven by the *BCL2* P1/P2 promoter region, under the control of the 3′ IgH superenhancer (under the form previously described as a functional “core 3′RR” [26], combining the various enhancer elements from the Ig heavy chain locus 3′ regulatory region, but devoid of the large intervening sequences located in-between enhancers in the mouse IgH locus);The “Igκ-*BCL2*” model, in which the Jκ region is deleted and replaced with the same above-mentioned *BCL2* cassette [P1/P2 promoter *BCL2* gene], as a knock-in in the mouse Igκ light chain locus, immediately upstream of the Eκ enhancer.

Both models thus contain the full promoter region of *BCL2*, including its two alternative promoters, P1 and P2, all in a mixed B6/129 genetic background.

### 2.2. Tumor Follow Up

A cohort of 19 Igκ-*BCL2* mice (12 homozygous and 7 heterozygous) and 8 3′RR-*BCL2* mice were monitored for the spontaneous development of tumors with time.

### 2.3. Flow Cytometry Analysis of Lymphoid Compartment

Single-cell suspensions from the BM and secondary lymphoid organs (spleen and mesenteric lymph nodes) were taken from mice that were either resting (unimmunized) or immunized with 3 iterative monthly SRBC injections. These cell suspensions were then labeled using various extracellular antibodies designed to identify the different early and late B-cell populations as well as the different T-cell populations in each of the organs mentioned above. Following the extracellular staining step, cells were fixed and permeabilized using the eBioscience FOXP3/transcription staining buffer set (Invitrogen, Reference: 00-5523-00) and then stained for intracellular proteins. The detailed list of antibodies used in each of these panels is summarized in Appendix A. Cells were then analyzed by flow cytometry using an LSR FORTESSA cytometer (Beckton Dickinson), and data analysis was performed using the Flow Logic system (the precise gating strategies for the different panels are listed in Appendix A).

Percentages were determined (for the different B cell sub-populations among CD19+ cells and for the different T cell sub-populations among CD4+ cells) as well as absolute cell numbers only in immunized mice (calculated based on the measured cell counts and lymphocyte percentages in the organ analyzed).

The mice used for evaluating lymphoid compartments represented 2 cohorts:Resting cohort: this cohort comprising 6 wildtype, 13 hemizygous Igκ-*BCL2*/+, and 7 3′RR-*BCL2* mice was analyzed to carry out full characterizations of mice at the resting state;Iteratively immunized cohort: 3 groups of 13 wildtype, 7 Igκ-*BCL2*/+, and 10 3′RR-*BCL2* mice were immunized intra-peritoneally at 3 months of age with 200 µL of Sheep Red Blood Cells (SRBC) and then iteratively over 3 consecutive months; they were sacrificed one month after the third immunization.

### 2.4. Western Blots

For the detection of human BCL2, we used wildtype mice as negative controls and the FL DoHH2 cell line as a positive control. Analyzed mouse samples included samples from transgenic (3′RR-*BCL2*) mice and from both homozygous (Igκ-*BCL2*/Igκ-*BCL2*) and hemizygous (Igκ-BCL2/+) Igκ knock-in mice.

A amount of 10 μg of total proteins were extracted (using 2× Laemmli buffer, Biorad, Catalog number # 161-0737, Composition: 65.8 mM Tris-HCl PH 6.8, 26.3% (*w/v*) glycerol, 2.1% SDS, 0.01% bromophenol blue) and denatured/reduced using β-mercaptoethanol (2.5% final) at 95 °C for 5 min.

After electrophoresis on a 12% polyacrylamide gel (Biorad, Hercules, CA, USA), proteins were transferred onto PVDF membranes (GE Healthcare). The human BCL2 protein was detected using the same mouse anti-human BCL2 antibody we use for flow cytometry, followed by an HRP-linked anti-mouse Ig antibody (eBioscience, San Diego, CA, USA), #18-8817-30).

Actin was stained as a housekeeping control protein with rabbit anti-actin antiserum (Sigma-Aldrich, Saint-Quentin, France A2066) followed by goat anti-rabbit-HRP (Southern Biotech, Birmingham, AL, USA, 4050-05).

Membranes were developed by enhanced chemiluminescence for a high-sensitivity detection system according to the manufacturer’s instructions (Bio-Rad, Marnes-ma-Coquette, France).

### 2.5. Proliferation Test

A separate cohort of 6 wildtype (WT), 4 Igκ-*BCL2*/+ hemizygous mice, and 5 3′RR-*BCL2* mice were injected intraperitoneally with 200 µL of SRBC at day 0, followed by another intraperitoneal injection with 200 µL of 5-ethynyl-2′-deoxyuridine (EdU) at Day 6, and they were sacrificed at day 7.

Single-cell suspensions were taken from the BM, spleen, and mesenteric lymph nodes and stained using the same panel described above but modified to be able to see the EDU+ cells. This was performed following the “Click-iT EdU Flow Cytometry Assay Kit protocol” (Molecular probes by Life Technologies, catalog numbers C10419, C10420). Cells were then analyzed by flow cytometry using the BD LSR FORTESSA, and the data analysis was performed using the Flow Logic system.

### 2.6. Quantification of Antibody Affinity

We followed the indirect enzyme-linked immunosorbent assay (ELISA) described by Zhang et al. to quantify the affinity of the produced antibodies (notably IgG) in our mouse models compared to wildtype.

Mice were injected with Ovalbumin (Ova) at 1 ng/mL and Addavax (equal volume for both) on Day 0 and Day 14. Sera were taken on day 28, and quantification of total IgG in the serum was first performed, followed by the ELISA-based affinity study. Briefly, 96-well plates were coated with Ova at 2 µg/mL. Day 28 sera at a fixed concentration of 5 × 10^−10^ M were incubated overnight with Ova at different concentrations but always in large excess compared to IgG (4 × 10^−7^ to 6.25 × 10^−9^ M). The next day, the mix of serum IgG/Ova was incubated in 96-well plates previously coated with Ova at 2 µg/mL overnight. Then, the alkaline phosphatase-conjugated secondary antibody (anti-Mouse IgG) was added. After 45 min of incubation, P-Nitrophenyl phosphate (1 mg/mL) was added, alkaline phosphatase activity was blocked with 3 M Sodium hydroxide (NaOH), and the optical density was measured at 405 nm using a Multiskan FC photometer.

Finally, the dissociation constant (K_D_) was evaluated by measuring the slope of the linear dependence [27].

### 2.7. Next-Generation Sequencing for Ig Repertoire Analysis and Global RNA Expression

We performed Ig repertoire sequencing analysis on RNA (500 ng–1.5 mg) extracted from:BM, spleen, and mesenteric LNs from the iteratively immunized cohorts, including 3 groups of non-tumoral mice: wildtype (9 mice), Igκ-*BCL2*/+ (5 mice), and 3′RR-*BCL2* (5 mice);Spleen, mesenteric LNs, and tumoral tissues from mice affected with tumors, including 8 Igκ-*BCL2* knock-in mice (5 Igκ-*BCL2* homozygous + 3 hemizygous), and the single 3′RR-*BCL2* transgenic mouse that developed lymphoma.

Repertoire sequencing was performed using RACE-PCR and high-throughput sequencing for determining cell repertoire diversity and the distribution of IgH clonotypes as previously described [28]. These experiments used a new generation methodology, which combines 5′ RACE PCR; sequencing; and, for analysis, the international ImMunoGeneTics information system (IMGT), IMGT/HighV-QUEST Web portal, and IMGT-ONTOLOGY concepts. Briefly, we amplified IgH transcripts with 5′ RACE PCR using a reverse primer. First, a mix containing up to 1 µg of RNs, 1 µL of reverse primers hybridizing within the Ig mu (µ) and gamma (γ) CH1 exons, and dNTP mix was incubated 3 min at 72 °C and 2 min at 42 °C. After a short spin, the mix was placed on ice for 2 min, and 1 µL of ProtoScriptII enzyme (New England Biolabs) was added, together with a cap-race primer (5′ AAGCAGTGGTATCAACGCAGAGTACAT[GGGG] 3′, where the 4 G between brackets are ribonucleotides). The resulting cDNA was amplified with Taq Phusion (New England Biolabs) using a universal forward primer mix (5′ CTAATACGACTCACTATAGGGC 3′ and 5′ CTAATACGACTCACTATAGGGCAAGCAGTGGTATCAACGCAGAGT 3′, in a ratio of 4:1), as described [16], and a mix of reverse primers hybridizing within the µ and γ CH1 exons. Cycling conditions were 30 s at 98 °C, 32 cycles of 30 sec at 98 °C, 30 s at 65 °C, 30 s at 72 °C, and final elongation 5 min at 72 °C. Illumina sequencing adapters and tag sequences were then added by primer extension using Taq Phusion. Cycling conditions were 30 s at 98 °C, 12 cycles of 30 sec at 98 °C, 30 sec at 65 °C, 30 sec at 72 °C, and final elongation 5 min at 72 °C. The resulting amplicons were sequenced on an Illumina MiSeq sequencing system using MiSeq Reagent kit V3 500 cycles. Paired reads were merged using FLASH [29]. Repertoire analysis was performed using IMGT/HIGHV-QUEST tool (http://imgt.org/ accessed on 17 October 2021) [30] and associated RStudio package scripts; associated tools are available on the IMGT Web site.

Global evaluation of transcriptomic profiles was obtained using RNA-seq on lymphoid tissue samples (spleen and mesenteric lymph nodes) from the non-tumoral, iteratively immunized cohort, comparing samples from animals with three different genotypes, either wildtype, Igκ-BCL2/+, or 3′RR-BCL2. Sequencing was performed on an Illumina MiSeq, in a 2 × 150 bp configuration. Read counts from RNAseq normalized using the featureCount script were compared with DESeq2 (version 1.26.0) for evaluating differential expression along the binomial negative distribution. The results provided by DESeq2 were then submitted to variance stabilizing transformation [31]. A matrix of normalized values restricted to genes with expression significantly modified (*p* < 0.01) in at least one genotype was then used in order to draw a heatmap of significantly modified signatures of interest using the “ggplot2” R package.

### 2.8. Single Cell Sequencing and Analysis

Three sets of 10-day splenic B cells were magnetically sorted (B cell negative selection kit, StemCell, Vancouver, Canada) from mice immunized once with SRBC. Single cells were captured and barcoded using the 10× 3′ sequencing kit (Chromium Next Gem Single cell 3′ reagent kit v3.1, 10× Genomics, Pleasanton, Ca, USA), and libraries were prepared following the manufacturer’s instructions. Libraries were run using 2 × 75 paired-end reads on the HiSeq4000 Illumina sequencer. Raw data were successively processed and analyzed with the 10× Cell Ranger (10× Genomics, Pleasanton, Ca, USA) and Seurat (v3.2.3) package [32]. Mean reads per cell were 40,766 for the WT sample, 66,971 for the 3′RR-BCL2 sample, and 102,596 for the Igκ-BCL2 sample. Median genes per cell over 3 samples varied from 1767 to 1869. Cells expressing less than 800 or more than 4500 genes or with more than 20,000 unique molecular identifiers (umi) counts were filtered out. Cells with a frequency of mitochondrial genes of more than 8% or with a frequency of ribosome genes of less than 10% were also removed from the analysis. Contaminating T cells and myeloid cells were detected based on canonical markers (Cd3 genes and C1q genes) and filtered out. Gene counts were normalized using the SCTransform package (v0.3.1) with a second non-regularized linear regression applied to the percentage of mitochondrial and ribosomal genes [33]. Canonical correlation analysis was used to integrate data from different batches by running the following steps as implemented in the Seurat package [34]: selection of integration features; removal of immunoglobulin genes from the features used for integration; preparation for integration with the PrepSCTIntegration function. Data integration was then performed using the FindIntegrationAnchors and IntegrateData functions based on the first 30 correlation components. PCA analysis was performed on the integrated dataset, and the first 12 principal components were used for UMAP computation and clustering (using 30 nearest neighbors and a resolution of 0.2). Cluster 1 was subclustered by executing the same steps described previously (PCA analysis on the subset and clustering computation with the first 10 principal components, 15 nearest neighbors, and a resolution of 0.2). The log-normalized expression values, marker genes for each cluster, as well as differentially expressed genes between conditions were inferred by the Wilcoxon test as implemented in the FindAllMarkers function. Cell cycle score and classification were calculated by CellCycleScoring function based on the expression of G2/M and S phase markers [35].

### 2.9. Study of BCL2 Promoter Mutations

For this study, DNA was phenol/chloroform extracted from:1.The various cell lines listed in Section 1 (DoHH2, SU-DHL-4, SU-DHL-6, and OCI-Ly3) were used as positive controls for BCL2 promoter mutations;2.Total spleen from iteratively immunized mice (4 Igκ-*BCL2* /+ and 4 3′RR-*BCL2*);3.Class-switched versus non-class-switched B cells from additional groups of 3-month-old immunized Igκ-*BCL2* /+ (3 mice) and 3′RR-*BCL2* mice (3 mice). These mice were induced only once with SRBC and sacrificed on day 7. Single-cell suspensions from the spleen were stained with BV510 anti-CD19 (BD Biosciences, Clone: 1D3) and FITC anti-IgM (Southern Biotech, Cat number: 1020-02) antibodies. Cells were then sorted using the BD Aria III cell sorter to obtain class-switched B cells (CD19^+^/IgM^-^) and non-class-switched B cells (CD19^+^/IgM^+^);4.From mice tumors:−Tumoral tissues of 3 Igκ-*BCL2/*+ mice;−Mesenteric LNs of one Igκ-*BCL2/*+ and one 3′RR-*BCL2* tumoral mice;−Total spleen of two 3′RR-*BCL2* tumoral mice.

The full promoter region of *BCL2* was then amplified by polymerase chain reaction (PCR) using specific forward (5′ TGAATGAACCGTGTGACGTTACGC 3′) and reverse (5′ CTCAGCCCAGACTCACATCA 3′) primers. The amplification of the PCR product (2,184 bp long) was verified by gel electrophoresis using agarose 2% gel in TBE. The amplified PCR product was then purified using the Nucleospin Gel and PCR Clean-up kit (Macherey-Nagel, Reference: 740609.250).

Next-generation sequencing was performed from amplified products (1 μg) according to the user guide of the Ion Xpress Plus gDNA Fragment Library Preparation (Life Technologies catalog no. 4471269), and libraries were sequenced on an Ion Proton System.

Two non-lymphoid controls were used for each library run, consisting of genomic tail DNA samples from an Igκ-*BCL2/*+ and a 3′RR-*BCL2* mouse, respectively. Analysis was performed using Deminer software developed by our laboratory in order to subtract the background level of mutations observed on the same sequence in a control experiment [36].

## 3. Results

### 3.1. Generation of Mice Carrying B-Cell Specific BCL2 Deregulation

Hemizygous mice corresponding to our two BCL2 deregulation models were studied. Both produced high amounts of a human BCL2 protein of normal size in a B-lineage-specific manner (Figure 1A–D). Both revealed overexpression of human BCL2 in spleen follicular and GC B cells, with a stronger expression in the Igκ-*BCL2* than in the 3′RR-*BCL2* mice (Figure 1D). In addition, BCL2 expression patterns along B-cell differentiation stages strongly differed between the two models (Figure 1D). The Igκ-locus driven deregulation consisted of very high and stable BCL2 overexpression throughout B-cell differentiation in the BM from B-cell progenitors to recirculating mature B cells, plasmablasts having migrated to the BM and mature plasma cells. This “knock-in” strategy also yielded a constitutively high expression in peripheral spleen and lymph node lymphoid tissues, homogeneously affecting all B-cell compartments but strongly culminating in plasmablasts.

In striking contrast, the IgH 3′RR-*BCL2* drove low BCL2 expression in BM B cell progenitors and plasma cells. In the spleen and lymph nodes, it provided the highest expression in GC B cells, either centrocytes or centroblasts, with expression remaining high without peaking in plasmablasts and falling down in plasma cells.

### 3.2. Impact of BCL2 Deregulation on B- and T-Cell Differentiation

We analyzed B cell compartments by flow cytometry in non-immunized mice bred in a SOPF facility, comparing mutant mice to WT controls. Prior to any immunization, the percentage of BM CD19+ cells was similar in BCL2-deregulated models to that in WT controls and showed no amplification of any progenitor compartment, even with lower percentages of pro-B and pre-B cells in Igκ-BCL2 mice (Figure 2A). In contrast, BM plasmablasts and PCs were constitutively increased in the Igκ-*BCL2* mice.

In the periphery of resting unimmunized mice, CD19+ cells were more abundant in Igκ-*BCL2* than in WT mice in both lymph nodes and spleen, while 3′RR-*BCL2* mice inconstantly showed such an increase. The difference between both strains was still more striking for plasmablasts and plasma cells, the amount of which was normal in 3′RR-*BCL2* mice but increased by up to ten-fold in Igκ-*BCL2* (Figure 2B,C). In contrast, marginal zone (MZ) cells were decreased in Igκ-*BCL2* mice. The ratio of CD93+CD138+ vs. CD93-CD138+ plasmablasts was rather increased in both strains, significantly in lymph nodes, suggesting an increased amount of recently differentiating plasmablasts [37]. In plasma cells, for which CD93 expression marks long-lived plasma cells (LLPCs) arising from T-dependent responses, there was a tendency to an increased ratio of such cells in the 3′RR-*BCL2* mice, but not reaching significance [37].

Regarding the GC B-cell compartments in the spleen and mesenteric lymph nodes (corresponding to background GCs developed in non-immunized mice), both strains showed a constitutive increase in both centroblasts and centrocytes and an increased ratio of centroblasts vs. centrocytes.

Relative to expanded B-cell populations, the percentage of helper T cells appeared significantly decreased in the periphery of Igκ-*BCL2* mice only (Figure 2D), with a tendency for both BCL2 strains to have less naïve T cells but slightly increased effector (Teff) and follicular helper T (Tfh) cell populations, only reaching statistical significance in the mesenteric LNs of the Igκ-*BCL2* mice (Figure 2E).

### 3.3. Lymphoid Compartments and Response to Immunization in BCL2 Mice

Human lymphoid malignancies are often correlated with past chronic activation by viral Ag or auto-Ag, and we thus explored the behavior of BCL2 transgenic mice in conditions of three consecutive B-cell stimulations with the particulate Ag SRBCs.

Such a repeated stimulation resulted in a strong global B-cell increase in all lymphoid tissues from Igκ-*BCL2* mice but not 3′RR-*BCL2* mice (Figure 3). The Igκ-*BCL2* also confirmed an MZ B-cell decrease.

In BM (Figure 3A), the B-cell increase corresponded to recirculating B-cells, whereas all B-cell progenitor compartments appeared to be decreased. Percentages and the absolute number of BM plasmablasts and plasma cells were also significantly increased in Igκ-BCL2 mice.

In spleen and lymph nodes from Igκ-*BCL2* mice (Figure 3B,C), the global increase in CD19+ B cells was also associated with a strong increase in plasmablasts and to a lower extent of plasma cells. Plasma cells quantified after immunization did not show the above-mentioned increased ratio of CD93+ cells and did not yet express the markers of LLPCs. In contrast, a strongly increased ratio of CD93+CD138+ plasmablasts was seen in the spleen of Igκ-BCL2 mice, indicating a high influx of recently differentiated cells [37].

Compared to Igκ-*BCL2* mice, 3′RR-*BCL2* mice showed a more specific relative increase in GC B-cells, involving both centrocytes and centroblasts, together with a higher ratio of centroblasts vs. centrocytes, suggesting that BCL2 expression in these mice mostly impacted B-cell survival at the centroblastic stage.

In order to evaluate whether the GC increase in both models and the plasmacytosis in Igκ-BCL2 mice rather involved the ongoing entry of B-cells into the GC and plasma cell stages or the accumulation of long-lived cells, we measured BrDU incorporation into GC B-cells and GC plasma cells one week after immunization (Figure 3D). Evaluation of BrdU incorporation in GC B cells showed a preserved ratio of recently divided vs. lately divided cells. In parallel, this experiment revealed a decreased ratio of BrDU+/ BrDU- plasma cells, indicating that in addition to the increased influx of plasmablastic cells, the accumulation of LLPCs also clearly contributed to peripheral plasmacytosis.

In order to further appreciate the functional impact of BCL2 expression on the ability of mice to mount an efficient immune response in the context of an altered GC regulation, we carried out immunization with the T-dependent antigen ovalbumin. We then quantified the affinity of anti-Ova IgG antibodies using an ELISA-based protocol for the calculation of the dissociation constant K_D_ (the higher the affinity of the produced antibody, the lower the K_D_). This protocol showed that both BCL2 models were able to produce a large amount of high affinity circulating IgG with a K_D_ similar to that obtained in WT controls (or even with a tendency towards a lower value, i.e., higher affinity) (Figure 3E). This finding is reminiscent of observations previously performed for Eµ-BCL2 mice, in which the selection of Ag-specific B-cells appeared to be maintained [38].

The relative representation of the various T-cell compartments after repetitive immune challenges with SRBC remained similar to those in resting mice, with a significant decrease in the peripheral helper T cell population in the Igκ-*BCL2* mice, together with increased Teff and Tfh cell populations in the LNs of Igκ-*BCL2* mice (and to a lesser extent in 3′RR-*BCL2* mice) (Figure 3F,G).

### 3.4. Early Impact of B-Cell Anomalies on Global Transcriptional Profiles from Lymphoid Tissue Populations

Beyond the abovementioned changes in B and T-cell populations shown by cell cytometry, we wished to evaluate the global impact of the B-lineage anomalies on lymphoid tissues. RNAseq evaluation of global transcriptomic profiles was thus carried out on a set of spleen and mesenteric lymph node samples (from the cohorts of mice iteratively immunized with SRBCs). Comparison of the cohorts of each type of BCL2-deregulated mice with WT controls showed significant changes in several typical signatures (Figure 4). Few differences were detected between WT and 3′RR-*BCL2* mRNA profiles, except for enrichment for a GC signature in the latter. This signature involved genes for multiple well-known markers of GC B cells, including the transcription factors *Mef2b* and *Mybl1*; genes encoding GC-specific enzymes such as *Aicda* (the activation-induced deaminase gene) and *Hpse* (Heparanase); and genes encoding signaling proteins known as up-regulated in the GC, such as *Nugggc* (encoding the Slip-GC nuclear GTPase), *Rassf6*, *Rgs13*, and finally *Efnb1* (encoding the membrane protein Ephrin-B1) [39,40,41,42]. The global transcriptional analysis of whole lymphoid tissues, in contrast, showed no increase in the plasma cell signature in 3′RR-*BCL2* samples.

A strong GC signature was also noticeable in Igκ-*BCL2* samples, especially in lymph nodes. However, in contrast to 3′RR-*BCL2* samples, Igκ-*BCL2* samples differed from WT for several additional signatures (Figure 4). The more prominent one was a strong plasma cell signature including constant Ig heavy and light chain genes but also *Jchain, Sdc1 (Syndecan 1), Bhlha15 (Mist1),* and *Tnfrsf17* (*BCMA*), which are all classical markers of plasmablasts and plasma cells [43,44,45]. Additional changes notably consisted of significantly down-regulated genes and thus did not correspond to signatures expressed by expanded B-cells but rather reflected a global impact of the B-cell and plasma cell expansion on other cells from the lymphoid tissue microenvironment. This included the down-regulation of many genes involved notably in TGFβ responses, but also in the inhibition of MAPK-dependent and of growth-factor-dependent responses. In addition, many myeloid-lineage specific genes, including *Klf6, CD83,Tagap, Fosl2,* and *Id2,* showed a clear down-regulation in Igκ-*BCL2* samples, while a few of them were up-regulated (such as *Bst2* and *Siglec-H*, two genes notably expressed in plasmacytoid dendritic cells) [46], altogether suggesting a change in the polarization of myeloid cells in the microenvironment of lymphoid tissues with BCL2-deregulated B cells (Figure 4).

### 3.5. Single-Cell Transcriptome Analysis

In order to appreciate the impact of BCL2 deregulation on the B-cell transcriptome more precisely than by global analysis of lymphoid tissues or by following a limited set of markers by cell cytometry, we carried out a single-cell analysis of purified B-lineage splenocytes 10 days after immunization by SRBCs. This analysis allowed identifying 12 clusters and sub-clusters. The top differentially expressed genes are indicated in Appendix A. From their transcriptional profiles, clusters were identified as transitional cells, recirculating resting B-cells, extrafollicular and interfollicular cells, IFN-activated cells, marginal zone, pre-centroblastic, centroblasts, early centrocytes (*Aid+ Bcl6+*), late centrocytes (Aid- Bcl6+), preplasmablasts, plasmablasts, and plasma cells (Figure 5A). This analysis confirmed and extended the observations made by flow cytometry. In particular, all compartments upstream of GC formation were rather decreased in BCL2 transgenics (from transitional cells to resting, extrafollicular, IFN-activated, marginal zone, and pre-centroblastic cells). In the Igκ-*BCL2* model, the amplified compartments began with centrocytes but then culminated with plasmablasts and plasma cells. In contrast, in the 3′RR-*BCL2* model, the amplified subclusters were more “GC-focused”, including centroblasts, centrocytes, late centrocytes, and plasmablasts (Figure 5B). This analysis also allowed for monitoring the mean expression level of a number of highly expressed genes comparatively between WT, Igκ-*BCL2,* and 3′RR-*BCL2* mice. With more precision than in flow cytometric measurements, it showed a broad overexpression of *BCL2* in all subclusters in the Igκ-*BCL2* model, compared with a lower and much more specific expression in 3′RR-*BCL2* B cells, focused on centroblasts, centrocytes, and plasma cells (Figure 5C).

### 3.6. B-Cell Diversity in Young Immunized Mice

Ig repertoires of 3-month-old immunized mice only showed minor changes in comparison to normal mice (Figure 6), without a significant increase in the Gini index in any of the Ig transcript categories analyzed (μ or γ IgH transcripts from spleen or lymph nodes), thus indicating at this stage that no clonal amplification was detectable but that normally diversified repertoires were expressed after immunization. The usage of VH subgroups associated with IgH µ or γ transcripts showed no significant variation compared to WT mice.

### 3.7. Immunohistochemical Analysis and Tumor Development in BCL2 Transgenic Mice

We allowed cohorts of 40 mice for each genotype to grow older in order to monitor the potential onset of spontaneous tumors. Aggressive tumors indeed started to appear at ages 41 to 117 weeks in 10 homozygous Igκ-*BCL2*/Igκ-*BCL2*, 9 heterozygous Igκ-*BCL2*/+ mice, and 6 3′RR-*BCL2* mice (Table 1 and Figure 7A).

Representative immunohistochemistry analyses of tissues either from young mice (in the pre-malignant stage) or old mice with overt tumor development (notably 3′RR-*BCL2* mouse Tg-6 and Igκ-*BCL2*/Igκ-*BCL2* mouse KI-8 in Table 1) are shown in Figure 7B.

At the pre-malignant stage, the spleen showed a broadly normal architecture but with large follicles densely occupied with CD19+ B cells mixed with less abundant plasma cells. In such young 3′RR-*BCL2* mice, BCL2 staining mostly marked B cells within GCs. On the contrary, in the Igκ-*BCL2* mice, abundant plasma cells already populated the red pulp at the early stage and stained strongly for BCL2.

Later, in mice carrying overt tumors from either genotype, the spleen red pulp appeared heavily infiltrated by CD138+ BCL2+ plasma cells. These tumors affected mostly the mesenteric lymph nodes, spleen, and liver. The tumors analyzed at other distant locations also consisted of CD138+ cells, and the disease thus appeared similar to a disseminated plasmablastic lymphoma (Figure 7B).

Ig repertoire analysis by REPseq was performed for tumors from four Igκ-*BCL2*/Igκ-*BCL2* tumoral mice (KI-2, KI-3, KI-4 and KI-8), three Igκ-*BCL2*/+ tumoral mice (KI-16, KI-18, and KI-19), and one 3′RR-*BCL2* tumoral mouse (Tg-6) (Table 2). All tumors analyzed in both strains included one or eventually two strongly predominant clonal cell populations, each so-called “predominant clonotype” representing 30% to 98% of all Ig reads (Table 2). In only two cases, the VDJ region included an N-glycosylation site, which was germinally encoded. Both IgM-producing and IgG clones were found, with five of them carrying no mutation of the expressed VDJ region, while seven showed some level of somatic hypermutation (from 1 to 21 mutations/Kbp), indicating that malignant clones could originate in some cases from an extra-follicular pathway or in other cases be GC-derived.

### 3.8. The BCL2 Cassette Is Exposed to Low SHM in Both Transgenic Models

Sequencing the promoter region of *BCL2* in both models revealed that SHM occurred all along a 1.5 kb sequence fragment and globally appeared diversified in polyclonal cells (in agreement with the B-cell diversity indicated by Ig repertoire experiments) in mice not affected with tumors (Figure 8). The mutation rate at a given position thus never exceeded 0.6%. Sequences with the highest rate of SHM were rather obtained in class-switched B cells from immunized Igκ-*BCL2* mice, and the most frequently mutated positions were located around the P1 promoter.

In some but not all tumors, mutations were present at a higher level, eventually approaching 10% of the reads.

## 4. Discussion

Deregulation of the BCL2 oncogene is observed in multiple different types of B-cell malignancies, with diverse patterns of expression, and the translocation of the oncogene to the IgH locus is the classical hallmark of FL [1]. This translocation and other alterations of the BCL2 locus architecture, such as either translocation to Ig light chain genes or chromosomal amplifications of the BCL2 locus, are also documented either in some rare FL cases, in DLBCL or in chronic lymphocytic leukemia cases [3,4,5]. In addition to MCL1-dependence, about 20% of myeloma cases, and notably those featuring t(11;14) translocations, are co-dependent from BCL2 deregulation and are thus responsive to BCL2-targeted therapy using venetoclax [47,48]. Altogether, diverse patterns of BCL2 deregulation clearly impact the phenotype of BCL2-driven B-cell malignancies in patients.

Regarding mouse models, the most popular BCL2 deregulation models are the Eµ-*BCL2* transgenics, with broad BCL2 deregulation at all B-cell stages, and the vavP-*BCL2* mice, with broad overexpression in both B and T cells [19,20,24]. We thus currently lack optimal mouse models for precisely reproducing the maturation stage-specific anomalies that support the development of human BCL2-driven B-cell malignancies.

We herein reported two new models of B-lineage-specific BCL2 deregulation in mice. These models yield different patterns of expression, either with a restricted GC-specific expression in transgenics driven by the IgH 3′RR or with a global BCL2 overexpression in the Igκ-*BCL2* knock-in strain, which affects all mature B-cell stages, including GC cells and reaches a climax in plasma cells. Although both models roughly yield deregulated expression in GC B cells, the high “GC-specificity” of the 3′RR-driven model makes it a potentially attractive “first hit” platform for studying early events of FL lymphomagenesis. Indeed, FL is currently considered as evolving from FLLC cells carrying an initial t(14;18) translocation then, followed by multiple additional driver mutations, the latter concurring to block plasma cell differentiation and the exit of the transformed cells from the GC stage.

In contrast, the broad deregulation and highest expression of the oncogene in both GC B-cells and plasmablasts observed after the Igκ-*BCL2* knock-in principally shows up with an early polyclonal expansion of the plasmablastic and plasma cell compartments in young mice, prior to any immunization. This plasmacytosis also involves both an increased influx of plasmablasts and the accumulation of long-lived plasma cells. In old animals, the pre-malignant plasmacytosis can then end with monoclonal or oligoclonal plasma cell tumors, and this mouse strain could thus provide a pertinent model for BCL2-driven plasma cell dyscrasia, which is notably a frequent feature of t(11;14) human myeloma cases [47,48].

Homoeostasis of immune cells in lymphoid organs is strongly dependent upon inter-cellular interactions, and both lymphoid and plasma cell malignancies are known to strongly involve interactions between B-cells and their microenvironment in lymphoid tissues [16,49]. In this regard, it is interesting to notice that the altered B-cell homeostasis in such B-cell-specific *BCL2* deregulation models has an early impact on non-B cells from the lymphoid microenvironment long before the development of overt B-cell malignancy. Notably, the imbalance of B-lineage compartments significantly alters T-cell populations, with lower amounts of naïve T cells but higher amounts of effector T cells and Tfh cells in the 3′RR-*BCL2* model than in WT animals, suggesting a global GC expansion, not restricted to GC B-cells. In contrast, in the Igκ-BCL2 mice developing polyclonal plasmacytosis, transcriptional analysis of lymphoid tissues shows not only a strong and expected over-expression of a plasma cell signature but also the significantly decreased expression of multiple genes, with signatures related to TGFβ signaling, IFN signaling, inhibition of MAPK signaling, and a strong down-regulation of many genes mostly specific for myeloid cells. It remains to be explored whether these early anomalies could pave the way for the future development of a suppressive tumoral microenvironment once aged mice develop plasma cell dyscrasias.

Finally, regarding the occurrence of spontaneous tumors, this preliminary characterization of mice solely affected with *BCL2* deregulation and no additional oncogenic hit shows that only some aged animals developed tumors. All consisted of plasma cell tumors, whatever the genotype, and whether or not the tumor was preceded by overt polyclonal plasmacytosis as seen in the Igκ-*BCL2* mice.

Mice from both genotypes were able to mount normal humoral responses, and the polyclonal expansion of some B-cell compartments was thus not associated with any differentiation blockade, which could have hampered the production of specific antibodies. A similar situation is likely occurring for human FLLCs only deregulating *BCL2*, which circulate and are reportedly identified in blood and lymphoid tissues from healthy individuals without any differentiation blockade [50,51]. In contrast, local amplification of a more aggressive B-cell clone, either in the early situation of in situ follicular neoplasia or of overt FL, is always associated with mutations additional to the BCL2 deregulation [52]. The most obvious roles of all such additional mutations, notably involving chromatin modifier genes such as CREBBP, EZH2, or KMT2D, is to reprogram B-cells towards an iterative GC reentry cycle by up-regulating BCL6 expression and inhibiting genes involved either into B-cell egress form the GC and/or into plasma cell differentiation and progress beyond the GC B-cell stage [53,54,55,56].

Although we have no indication about the eventual supplemental hits supporting the growth of plasmacytoma in aged *BCL2* mice from our current study, acquired genetic or epigenetic anomalies are likely since no tumor arises in young mice, whatever the *BCL2* expression dosage. In both mouse strains, B-cell repertoire diversity in young tumor-free animals appears normal, without monoclonal or oligoclonal pre-malignant expansion. Spontaneous and random genetic anomalies accounting for the development of plasma cell malignancies in older animals then clearly did not block plasma cell differentiation and/or egress from the GC. As for previously reported BCL2 deregulation models, the development of lymphoma in such mice will thus need on-purpose breeding with mouse strains carrying the same types of second hits seen in human GC lymphomas [16,53,54,55,56].

Contrary to our expectations, the presence in those mouse models of a complete P1/P2 BCL2 promoter fragment, including the negative regulatory sites for BCL6 binding, does not repress BCL2 expression in GC B-cells strongly express the mouse transcription factor Bcl6. This promoter is not affected either by obviously acquired mutations unleashing expression in some B-cell clones. Although the P1/P2 fragment is accessible to a significant amount of SHM, notably when inserted in the Igκ locus, these mutations rather seem to accumulate randomly at a low level. The spontaneously high expression of BCL2 in either model appears sufficiently high by itself for supporting the outgrowth of clonal malignant tumors without the need for promoter mutations.

Tumors occurring in both models are mostly made up of clonal plasma cells (with biclonal proliferations in some cases), either producing IgG or IgM with roughly similar frequencies and revealing SHM of VH genes ranging from 3.5 to 21 per Kbp for 7 out of 12 malignant clones, thus with a clear indication of a “post-GC” origin of the malignant clones. GC-derived plasma cells have indeed been described with a strongly higher SHM rate in comparison to extrafollicular plasma cells (mean 5.6 mutations/Kbp vs. 0.7/Kbp) [57]. In 5 out of 12 malignant clones studied, completely germline Ig sequences were expressed by the monoclonal B-cell clone, the origin of which was thus rather extra-follicular and independent of T-cell help.

Although attributed to V(D)J recombination errors, which could potentially affect any Ig locus, BCL2 translocations associated with FL in patients overwhelmingly involve the IgH locus on chromosome 14 and much more rarely Ig light chain loci [3,58]. Such translocations replace all the IgH V gene clusters with BCL2 and thus associate the oncogene with regulatory elements for the IgH constant gene cluster, hereby predominantly placing the oncogene under the control of the IgH 3′RR in mature B-cells. Such conditions yield strong BCL2 expression in activated B-cells and at the GC stage undergoing 3′RR-dependent SHM and CSR, as reproduced in our 3′RR-*BCL2* mouse model.

That both BCL2 strains from this study differ for the polyclonal plasma cell expansion developed in Igκ-BCL2 mice reveals an unexpected impact of the BCL2 level on B-cell homeostasis. The ability of B-cells to mount high-affinity Ag-dependent responses after immunization is, however, preserved in both *BCL2* models. Since the expansion of GC B-cells (in 3′RR-*BCL2* mice) and the expansion of GC and plasma cells (in Igκ-*BCL2* mice) does not occur at the expense of other B cell compartments, they are unlikely to result from impacts on cell fate decisions, for example making Igκ-*BCL2* cells more prone to engage into plasma cell rather than memory B-cell differentiation when BCL2 exceeds a certain threshold.

The simplest explanation for the observed expansion of some B-cell compartments is rather a lower level of cell death within GCs for both models, then increasing the influx of all GC and post-GC cells. Specifically in the Igκ-*BCL2* model, the prolongation of BCL2 overexpression beyond the GC or extra-GC B-cell activation stages then additionally increases the half-life of cells transiting through the plasmablast stage and finally supports the accumulation of LLPCs. Initial steps of plasma cell differentiation are indeed known to depend on endogenous BCL2 expression [59], and survival thus initially increases when BCL2 expression fails to tamper in transgenic Igκ-*BCL2* plasmablasts. Later on, LLPCs normally do not rely on BCL2 for their survival but rather on BCMA-dependent MCL1 expression [59]. Since MCL1 and BCL2 share a similar pro-survival function, maintaining or even increasing expression of the knock-in Igκ-BCL2 in all PCs necessarily ends with deregulated survival and hypertrophy of the LLPC compartment. Plasma cells are also strongly exposed to endothelial reticulum (ER)-mediated stress, which triggers autophagy and mitophagy. The higher Igκ-*BCL2* expression might then reach a threshold for which non-classical functions of BCL2 can be heightened beyond its main role in mitochondrial outer membrane permeability. BCL2-family members such as BCL2-L13 and BNIP3 are indeed also major actors of autophagy and mitophagy [60,61]. These factors share interactions with some BCL2 partners, such as BECLIN-1, and their function in promoting mitophagy might likely be hampered by excessive amounts of BCL2. Noticeably, human myeloma often associates up-regulation of MCL1 and silencing of BNIP3 [62]. Altogether, either by reducing ER-stress-induced apoptosis, autophagy, or mitophagy, the BCL2 up-regulation triggered at the plasmablast/plasma cell stage in Igκ-*BCL2* mice appears to open up the gate for entry of more B cells into the plasma cell stage together with deregulated accumulation of non-dying LLPCs. This model hereby appears as ideally suited for yielding a permanent polyclonal expansion of plasma cells and then provides an ideal platform for studying the second hits, which more profoundly affect homeostasis and support the development of plasma cell dyscrasias.

## 5. Conclusions

In comparison to the most widely used vavP-BCL2 model, the two models reported in this study are of high interest due to their complete B-cell specificity, while in contrast, vavP-BCL2 mice largely express BCL2 in other hematopoietic lineages and notably T-cells [24]. Interestingly, our study shows that even B-cell specific deregulation can indirectly affect the other cell populations present in lymphoid tissues, showing a B-cell impact on the lymphoid micro-environment even in pre-malignant conditions and then providing ideal conditions for studying the evolution of cell interactions along lymphomagenesis.

At this stage, our study of mice carrying a single on-purpose mutation obviously carries limitations. In contrast, spontaneous human lymphomas often combine multiple oncogenic anomalies [63]. We indeed compared two different locations of the same oncogenic cassette, but various parameters differ in these two architectures. Our main guess is that the IgH 3′RR is providing a higher specificity of expression for those stages corresponding to activated B-cells and GC B-cells and could thus provide optimal pre-malignant settings for lymphomas of the GC or post-GC type. It is indeed noticeable that the 3′RR not only includes transcriptional enhancers but also sites for transcriptional repression after binding factors such as Bach2, Mafk, and Pax5 [64,65,66,67]. In contrast, expression of a knock-in oncogene within the Igk locus yields expression at all B-cell stages under the influence of both the intronic Ek and the downstream kE3′ enhancer. By analogy, the Igk chain expression, BCL2 expression, is then the highest in plasma cells. However, stage-specificity is not the sole divergent parameter between both models. At all stages, the BCL2 transcription level is also higher for the Igk-BCL2 model. Chromatin marks could also differ since the 3′RR is known to recruit specific epigenetic marks, and the same is true for nuclear location and genome architecture since the 3′RR superenhancer was shown to play a role in long-distance interactions and chromosomal loop extrusion in activated B cells [68,69]. It remains to be determined which of these multiple parameters most strongly impacts the expression of a transgenic or a translocated oncogene.

## Figures and Tables

**Figure 1 cancers-14-05337-f001:**
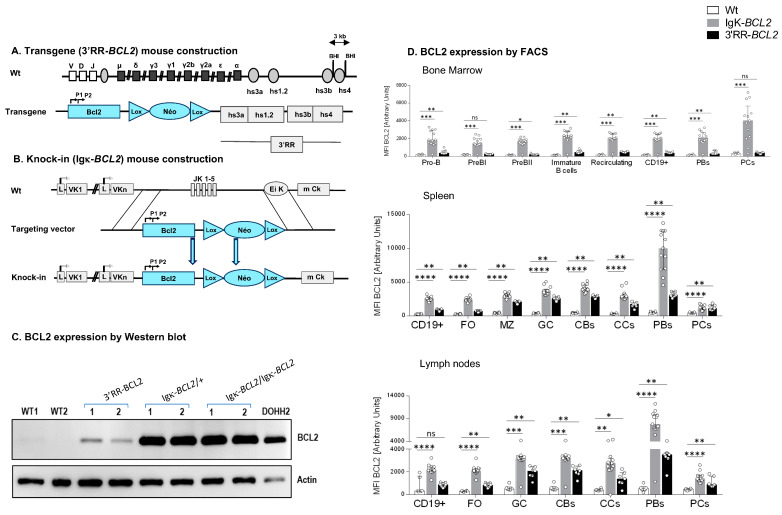
Mouse model construction schematics for (**A**) 3′RR-BCL2 transgenics and (**B**) Igκ-BCL2 knock-in mice. (**C**) Western-blot evaluation of human BCL2 protein expression in the spleen of WT (negative control), 3′RR-BCL2, Igκ-BCL2 ∆/+, Igκ-BCL2 ∆/∆ mice, and in the DoHH2 human cell line (positive control). (**D**) Human BCL2 expression in both mouse models, using the mean fluorescent intensity (MFI) obtained by flow cytometry from bone marrow, spleen, and mesenteric lymph nodes B-cells from unimmunized 3′RR-*BCL2* and Igκ-*BCL2* ∆/+ only. (* *p* < 0.05; ** *p* < 0.01; *** *p* < 0.001; **** *p* < 0.0001).

**Figure 2 cancers-14-05337-f002:**
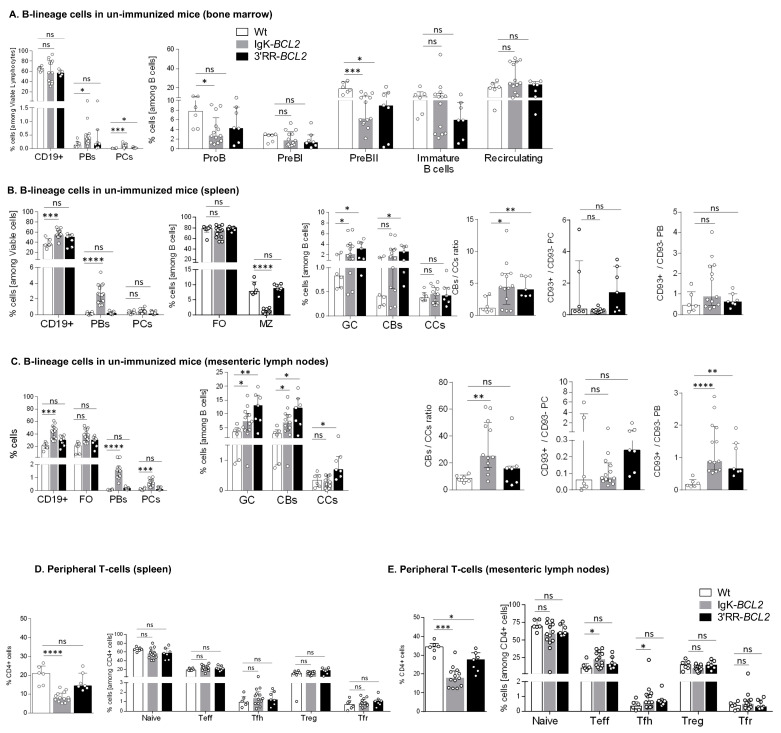
B cell development and lymphoid compartments in resting mice Flow cytometry data showing the impact of BCL2 deregulation alone, in both mouse models compared to WT mice in non-immunized conditions, on the different B cell compartments, in bone marrow (**A**), spleen (**B**) and mesenteric lymph nodes (**C**), as well as on the T cell compartment in spleen (**D**) and mesenteric lymph nodes (**E**). Percentages of the different cell compartments are presented as medians with interquartile range and the statistical significance was determined by the Mann–Whitney U test. Ratios for the centroblasts to centrocytes, as well as CD93+ to CD93− plasma cell (CD138+ CD19−) and plasmablastic cell (CD138+ CD19+) populations, are also presented (* *p* < 0.05; ** *p* < 0.01; *** *p* < 0.001; **** *p* < 0.0001). The full staining panel of both B and T cells is summarized in Appendix A.

**Figure 3 cancers-14-05337-f003:**
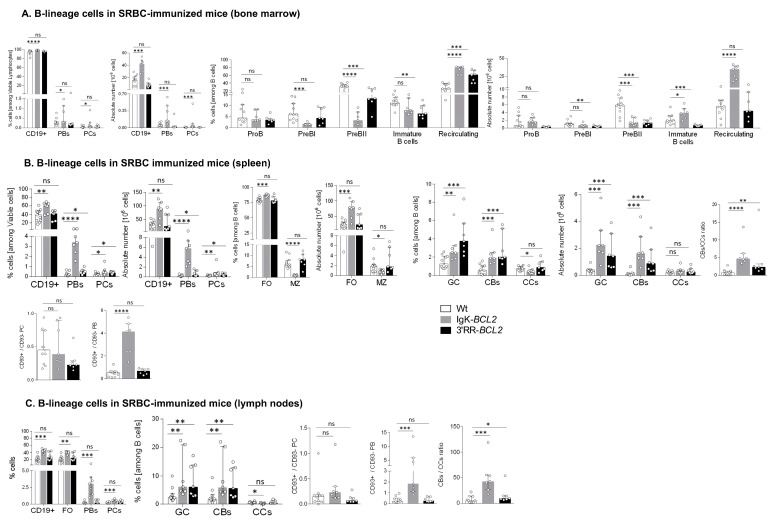
Immune responses in immunized mice (**A**–**C**) Impact of BCL2 deregulation on the immune response of immunized mice lymphoid tissues (bone marrow, spleen, and mesenteric lymph nodes, respectively), of intra-peritoneal immunization (3 iterative SRBC injections, on a monthly basis). B cells were evaluated by flow cytometry in BCL2 mice compared to WT. The ratios of centroblasts to centrocytes, as well as CD93+ / CD93- plasma cells and plasmablasts, are indicated. Percentages of the different populations, as well as absolute cell numbers (calculated based on the percentages obtained by flow cytometry and the total amount of lymphocytes in each organ), are presented as medians with interquartile range: statistical significance was determined using the Mann–Whitney U test. The full staining panel and gating strategy are summarized in Appendix A. (**D**) Flow cytometry analysis of splenocytes stained with EDU (24 h post intraperitoneal EDU injection and 7 days after SRBC immunization) in WT (5), Igκ-BCL2 ∆/+ (4) and 3′RR-BCL2 mice (5). The percentages of EDU-stained plasma cells (EDU+ PC) and GC B cells (EDU+ GC) are medians with interquartile range. Statistical significance was determined using Mann–Whitney U test. EDU staining was revealed by coupling to Alexa 488, together with specific markers for plasma cells (CD138 BV786), GC B cells (GL7-APC and CD38-APC-R700), and cell viability (FVS 780). (**E**) Affinity of anti-Ova antibodies. Histogram of the dissociation constant (KD) of IgG to Ova, evaluated by ELISA, in the sera of BCL2 mice compared to WT. Lower KD marks a higher affinity for Ova. KD values are medians with interquartile range. (**F**,**G**) Flow cytometry data showing the impact of the BCL2 deregulation accompanied with 3 iterative intra-peritoneal SRBC injections (on a monthly basis) on the different T cell populations, in both BCL2 mouse models compared to WT, in the periphery (spleen and LNs). Both percentages and absolute cell numbers of each T cell population are presented as medians with interquartile range, and the statistical significance is realized via the Mann–Whitney U test. (* *p* < 0.05; ** *p* < 0.01; *** *p* < 0.001; **** *p* < 0.0001).

**Figure 4 cancers-14-05337-f004:**
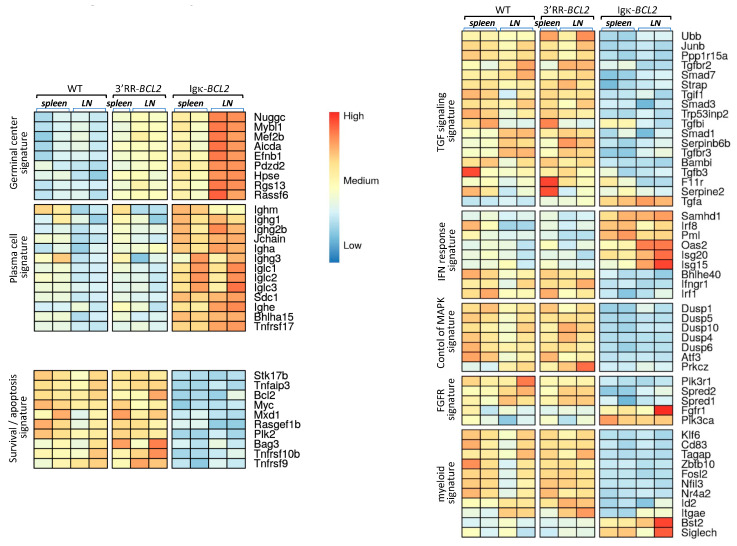
Transcriptomic signatures in whole lymphoid tissues from SRBC-immunized animals. Main transcriptional signatures, combining genes with significant variations, comparing WT to 3′RR-BCL2 and Igκ-BCL2 RNAseq profiles from lymphoid tissues (spleen and mesenteric lymph nodes) taken from SRBC-immunized mice.

**Figure 5 cancers-14-05337-f005:**
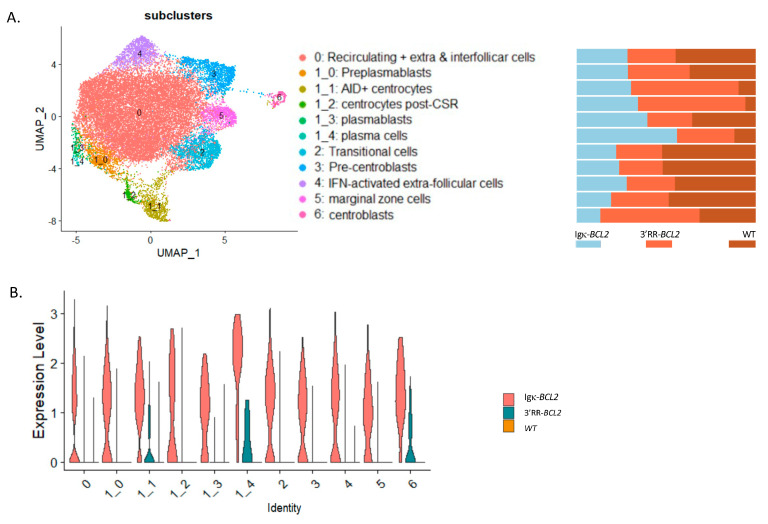
Single-cell RNAseq analysis of splenocytes from immunized mice: (**A**, *left panel*) UMAP of the 10× single cell analysis of spleen B-cells 10 days after SRBC immunization, split into 12 clusters (as defined in Appendix A). (**A**, *right panel*) Quantitative variations in cell numbers in the various clusters in Igκ-BCL2 and 3′RR-BCL2 mice compared to WT. (**B**) Human BCL2 expression level in the various cell clusters in Igκ-BCL2 and 3′RR-BCL2.

**Figure 6 cancers-14-05337-f006:**
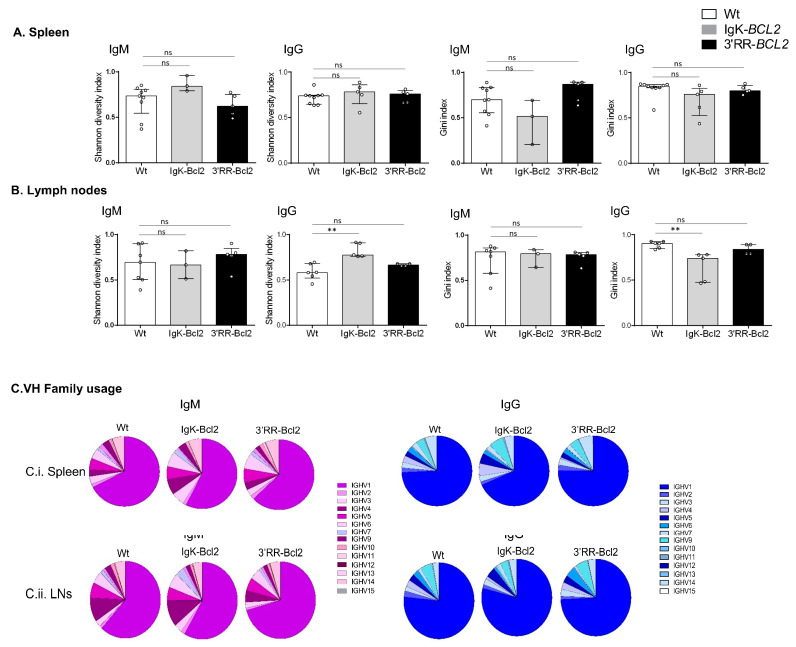
IgH repertoire of immunized mice. Histograms showing the Shannon diversity index and the Gini index of both IgM and IgG repertoires in the spleen (**A**) and mesenteric LNs (**B**) of both BCL2 mouse models compared to WT. Both indices are presented as medians with interquartile range, and statistical significances are determined by Mann–Whitney U test. (**C**) Pie charts showing the mean percentage of each VH gene family usage of both IgM (in purple) and IgG (in blue) in the spleen and mesenteric LNs of both BCL2 mouse models compared to WT. ** *p* < 0.01; ns, not significant.

**Figure 7 cancers-14-05337-f007:**
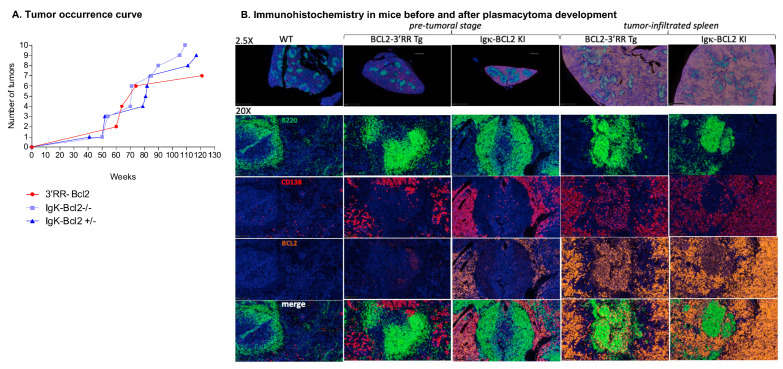
Tumors in BCL2 mice: (**A**) Curve showing the evolution of tumor occurrence in both BCL2 models with respect to their age (in weeks). (**B**) Immunohistochemistry (IHC) analysis of the spleen at the pre-tumoral and tumor-infiltrated stage of both BCL2 models compared to WT. Staining was performed for B cells (using B220 in Fitc ⟶ Green), plasma cells (CD138 in Tritc ⟶ Red), human BCL2 (in cyanine 5 ⟶ Yellow), and finally, DAPI in blue.

**Figure 8 cancers-14-05337-f008:**
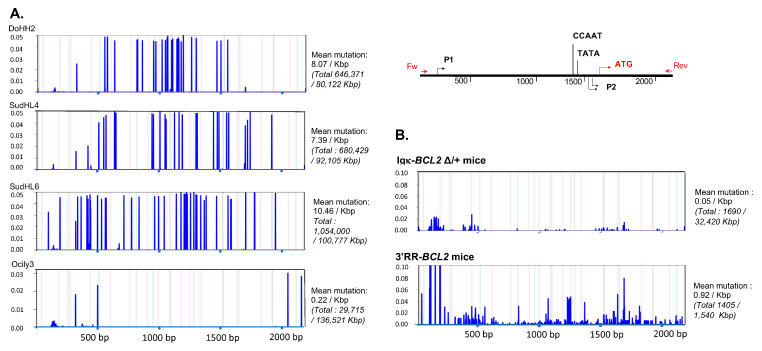
Mutation in the Bcl2 promoter region. Graphs showing the frequency and the localization/distribution of mutations along the BCL2 promoter region (after high-throughput sequencing) as well as the global mutation rate and the mutation enrichment in AID-hotspots in several human DLBCL cell lines as positive controls (**A**), in total splenocytes or class-switched versus non-class-switched splenocytes of immunized mice (3 iterative immunizations or single 7-day immunization, respectively) from both BCL2 models (**B**) and finally in various tissues/organs from several tumoral mice of both BCL2 models (**C**). Graphs are shown as generated by the system Deminer after correction with respect to two negative controls passed on the same run.

**Table 1 cancers-14-05337-t001:** Description of tumors: Table describing the mice that developed tumors in both BCL2 models. This cohort comprised 19 Igκ-BCL2 (total), and 8 3′RR-BCL2 mice were allowed to grow older, and these tumors were monitored for tumor development over time. For each mouse, the age at which they developed the tumor (or pre-tumoral symptoms), along with a detailed description of the location and the phenotype of each tumor, is listed.

Genotype	Number	Age (weeks)	Ascitis	Tumor	Organs Affected / Tumor Description	Other Observations
IgK-Bcl2 Δ/Δ	KI- 1	85	✓	✓	Intestine, Mesenteric LNs, Cervical LNs, Splenomegaly (165 mg)	/
	KI- 2	50	X	✓	Intestine, Mesenteric LNs, Cervical LNs, Splenomegaly (650 mg)	/
	KI- 3	71	✓	✓	Intestine, Stomach, Splenomegaly (395 mg)	/
	KI- 4	90	X	✓	Intestine, LNs	occlusion
	KI- 5	71	X	✓	Mesenteric LNs, Liver, Splenomegaly (1580 mg)	/
	KI- 6	54	✓	✓	Stomach, Intestine, Mesenteric LNs, Liver, Pancreas, Splenomegaly (305 mg)	/
	KI- 7	54	✓	✓	Kidneys, LN atrophy, Splenomegaly (950 mg)	/
	KI- 8	105	X	✓	Mesentery, Liver, Kidneys, Splenomegaly	/
	KI- 9	109	✓	✓	Liver, Splenomegaly (397 mg)	/
	KI- 10	70	X	✓	Stomach, Liver, Splenomegaly (287 mg)	/
KI BCl2 Δ/+	KI- 11	92	✓	✓	Under lungs, Mesenteric LNs, Splenomegaly (145 mg)	Diffused and bloody cervical LNs
	KI- 12	84	X	✓	Splenomegaly, Liver, Kidneys	Swollen testicles
	KI-13	117	✓	✓	Intestine, Atrophy of all LNs, Splenomegaly (2500 mg)	/
	KI-14	52	X	✓	Atrophy of cervical LNs, Splenomegaly (300 mg)	Obese
	KI-15	52	X	✓	Splenomegaly (300 mg)	Obese
	KI-16	41	X	✓	Mesentery, Splenomegaly	/
	KI-17	81	X	✓	Mesentery, Splenomegaly	/
	KI-18	111	X	✓	Mesentery, Splenomegaly	/
	KI-19	79	✓	✓	Mesentery, splenomegaly (1350 mg)	Decolored liver
3′RR-Bcl2+	Tg-1	74	X	✓	Atrophy of inguinal and axillary LNs, Splenomegaly (380)	/
	Tg-2	74	X	✓	Splenomegaly (161 mg)	/
	Tg-3	60	X	✓	Splenomegaly (240 mg)	/
	Tg-4	60	X	✓	Splenomegaly (222 mg)	/
	Tg-5	64	X	✓	Atrophy of cervical LNs	/
					Splenomegaly (222 mg)	
	Tg-6	64	✓	✓	Intestine, Liver, Atrophy of cervical LNs, Splenomegaly (730 mg)	/
	Tg-7	121	X	✓	Mesentery, Splenomegaly (1139 mg)	/

**Table 2 cancers-14-05337-t002:** Mutation in the expressed V region of mouse tumor samples. Table showing the detailed IgM and IgG repertoire analyses of the tumor tissues of mice from both BCL2 models. Total number of clones for each heavy chain and the corresponding number of productive reads are indicated. Based on the previous repertoire data obtained from WT mice, a certain threshold for the clone frequency was used, and only the clones that were present at a frequency higher than or equal to 30% were selected. For each clone, the V and J genes used along with their corresponding V mutation frequency (without CDR3 as it is highly mutated) are also indicated.

Genotype	Mouse nb	Heavy Chain	Total Number of Clonotypes	Total Number of Productive Reads	V_Gene	J_Gene	Clonotypic Predominance	V_Mutation Frequency
							(Clones ≥ 30%)	(without CDR3) per kb
IgK-BCL2 ∆/∆	KI-2	IgG	234	1269	IGHV4-1	IGHJ2	42.2	1
	KI-3	IgM	248	9087	IGHV6-3	IGHJ2	95.2	21
		IgG	58	8308	IGHV10-3	IGHJ2	99	3.5
	KI-4	IgM	430	2104	IGHV4-1	IGHJ2	47.6	0
		IgG	1055	6274	IGHV1-5	IGHJ3	30.2	15
	KI-8	IgM	143	1560	IGHV3-6	IGHJ2	64.4	18
		IgG	26	2835	IGHV4-1	IGHJ2	98.4	8
IgK-BCL2 ∆/+	KI-16	IgG	15	871	IGHV10-1	IGHJ2	98.8	0
	KI-18	IgG	74	175	IGHV10-3	IGHJ2	41.1	3.5
	KI-19	IgM	494	2106	IGHV1-39	IGHJ2	53.4	0
		IgG	619	11924	IGHV1-26	IGHJ2	85.2	0
3′RR-BCL2	Tg-6	IgG	1157	12395	IGHV2-2	IGHJ4	51.1	0

Clones carrying a VDJ N-glycosylation site are in red.

## Data Availability

All next-generation sequencing data are deposited on G.E.O. (details information can be found in Appendix A).

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
