# Peer review of "Distinct B-Cell Specific Transcriptional Contexts of the BCL2 Oncogene Impact Pre-Malignant Development in Mouse Models"

_cancers, 2022, doi:10.3390/cancers14215337_

Round 1

Reviewer 1 Report

This paper mainly focuses on the transcriptional contexts of the BCL2 impact pre-malignant development in two mouse models.  The idea of the article is clear and the method is proper, and the result is expounded accurately. However, a few minor points should be addressed before publication.

1. In the Introduction, a brief description of what has been said is enough without elaboration in the last paragraph.

2. In the Materials and Methods, the next-generation sequencing part should be re-described, as sequencing depth, sequencing data, etc. 

3. In the Discussion, what is the limitation in this paper? That should be pointed out.

Reviewer 2 Report

This manuscript by Zawil et al. addresses an important question regarding the role of different modes of BCL2 overexpression in the formation of B-cell malignancies, using transgenic mouse models. A model mimicking follicular lymphoma, where BCL2 is associated with the IgH locus 3’RR superenhancer, yields expansion of BCL2-high GCB cells. A knock-in of BCL2 within the Igk-locus yields the highest expression in plasmablasts and plasma cells, promoting their expansion and accumulation. The methodology employed by this study is appropriate and the results are presented in a clear way.

Major comments:

·         After reading the manuscript, I am still not sure why Igk and IgH -driven BCL2 expression has slightly different phenotypic outcome. Do the authors have a hypothesis that they could discuss in the manuscript?

·         I am not convinced that IGH-superenhancer-dependent BCL2 upregulation really recapitulates follicular lymphoma biology here, since both models lead to plasmacytoma development. Both transgenic models seem to lead to the same – expansion of plasmablasts and plasma cells.

·         Do the authors have a confirmation that this expansion is monoclonal or polyclonal?

·         Discussing the results in relation to other BCL2-OE models, such as CD19-Cre, VavP-, AID-Cre could be helpful to establish similarities and differences characterizing the new models described in this work.

Minor comments:

·         Line 70: pre-malignant

·         The manuscript would greatly benefit from a graphical abstract summarizing the results.

Round 2

Reviewer 2 Report

The authors answered all my questions during the review. I recommend publication of the manuscript in the current form in Cancers. Congratulations to the authors.